# A Multidisciplinary Journey towards Bone Tissue Engineering

**DOI:** 10.3390/ma14174896

**Published:** 2021-08-28

**Authors:** Sara G. Pedrero, Pilar Llamas-Sillero, Juana Serrano-López

**Affiliations:** 1Experimental Hematology Lab, IIS-Fundación Jiménez Díaz, UAM, 28040 Madrid, Spain; sara.gpedrero@quironsalud.es (S.G.P.); pllamas@fjd.es (P.L.-S.); 2Hematology Department, Fundación Jiménez Díaz University Hospital, 28040 Madrid, Spain

**Keywords:** bone tissue engineering, bone remodeling, bone disorders, biomechanics, scaffolds, microenvironment, 3D bioprinting, computational modeling

## Abstract

Millions of patients suffer yearly from bone fractures and disorders such as osteoporosis or cancer, which constitute the most common causes of severe long-term pain and physical disabilities. The intrinsic capacity of bone to repair the damaged bone allows normal healing of most small bone injuries. However, larger bone defects or more complex diseases require additional stimulation to fully heal. In this context, the traditional routes to address bone disorders present several associated drawbacks concerning their efficacy and cost-effectiveness. Thus, alternative therapies become necessary to overcome these limitations. In recent decades, bone tissue engineering has emerged as a promising interdisciplinary strategy to mimic environments specifically designed to facilitate bone tissue regeneration. Approaches developed to date aim at three essential factors: osteoconductive scaffolds, osteoinduction through growth factors, and cells with osteogenic capability. This review addresses the biological basis of bone and its remodeling process, providing an overview of the bone tissue engineering strategies developed to date and describing the mechanisms that underlie cell–biomaterial interactions.

## 1. Introduction

Bone is a dynamic connective tissue that plays a crucial role in locomotion, mechanical support and protection of soft tissues, calcium and phosphate storage, and harboring of bone marrow [1,2].

Bone presents the intrinsic capacity for modeling and remodeling in order to preserve skeletal size, shape, and structural integrity [3]. Bone modeling involves the coordination of bone formation and resorption, being related to bone growth and mechanical adaption. Bone remodeling is responsible for removal and repair of damaged bone, necessary for fracture healing, stimuli adaptation, and homeostasis [1,4]. The proper coupling of these two processes allows normal healing of most small bone injuries [2]. However, larger bone defects and diseases require additional stimulation to fully regenerate and heal. In this context, millions of patients suffer yearly from bone fractures, mainly caused by accidental fractures, aging-related disorders, and autoimmune diseases [5]. Indeed, due to the growing prolonged life expectancy, there has been a rapid increase in musculoskeletal pathologies such as scoliosis, osteoporosis, bone tumors, congenital defects, and rheumatic diseases such as osteoarthritis [6].

The existing routes to address bone disorders include medical procedures, transplantation, and medication [5]. Among them, gold standard grafting methods and drugs present drawbacks concerning their efficacy and cost-effectiveness [4], and numerous side effects and long-term consequences in the normal bone remodeling [7], respectively. Thus, alternative therapies become necessary to overcome these limitations.

In this context, bone tissue engineering (BTE) emerged in recent decades as a promising strategy for treatment of bone pathologies. BTE aims to combine engineering and biological properties to generate temporary artificial environments specifically designed to facilitate bone tissue growth [8,9]. The principal strategies developed to date include bioactive scaffolds, nanomedicine, cell-based combination products, and 3D printed bioceramics, among others [10]. Overall, great efforts have been made towards tissue engineering strategies that can meet three essential factors: osteoconductive scaffolds, osteoinduction through growth factors, and cells with osteogenic capability [11].

Regulation of 3D scaffold properties allow the selective guidance of cell development and differentiation in bone tissue. Moreover, computational modeling has emerged as an excellent tool to predict and optimize the clinical potential in terms of cell proliferation and differentiation, tissue growth, adaptation, and maintenance [12]. Thus, a proper understanding of the molecular mechanisms involved in cell–biomaterial interactions will allow one step forward in the achievement of functional advanced biomaterials for tissue engineering (Figure 1).

Herein, the basis of bone biology and the remodeling process are examined, providing an overview of the BTE strategies developed to date and the biological phenomena that underlie cell–biomaterial interactions. Moreover, computational modeling approaches for BTE are examined as a potential tool to help design better biomimetic materials.

## 2. Bone Biology

### 2.1. Bone Composition

Bone is a dynamic tissue that plays multiple functions in the body, including mechanical (structural support, locomotion, and protection of vital structures), and metabolic functions (mineral and acid–base homeostasis, calcium and phosphate storage, and harboring of bone marrow) [1,10,13]. Bone consists of a biomineral medium where four types of living bone cells are embedded: osteoblasts, osteocytes, osteoclasts, and bone-lining cells, which together constitute the basic multicellular unit (BMU) [1].

#### 2.1.1. Osteoblasts

Osteoblasts are specialized bone-forming cells that exert an important role in bone remodeling through the synthesis, deposition, and mineralization of the bone matrix [3,10]. They create an extensive network of intercellular communication together with their progenitors, which play a functional role in beginning and directing new bone formation [13,14]. Moreover, numerous transmembrane proteins such as integrins, connexins, and cadherins and specific receptors are responsible of anchorage regulation, which has a direct impact on cellular function in their response to metabolic and mechanical stimuli [14].

Osteoblasts develop from mesenchymal cells (MSCs) in bone marrow and other connective tissues. These cells can differentiate through different pathways, leading to adipocytes, myocytes, chondrocytes, and osteoblasts, depending on regulatory transcription factors [3,15]. Particularly, synthesis of bone morphogenetic proteins (BMPs) and Wnt signaling pathways are crucial in regulating osteoblast differentiation and function [15,16]. When activated, they can follow different paths: remain quiescent osteoblasts or bone-lining cells, become osteocytes, or return to the osteoprogenitor cell line [13].

#### 2.1.2. Osteocytes

Osteocytes constitute 90–95% of total bone cells; they are terminally differentiated osteoblasts embedded and immersed in the bone matrix [3,17]. They acquire dendritic morphology [1,15], which results in an elaborate osteocyte network that extends through the channels in the bone matrix, interacting with other osteocytes or osteoblasts from the bone surface [15,16]. This network allows osteocyte vascular supply, intercellular transport of small signaling molecules, and, in general, the communication among bone cells. Moreover, through this interconnected system, osteocytes act as mechanosensory cells, detecting mechanical strains and associated bone microdamage [3], thereby helping coordinate adaptive skeletal changes in response to mechanical loading [1,18]. For this reason, osteocytes are thought to initiate the bone remodeling process by a piezoelectric effect, which involves the conversion of mechanical stimuli into biochemical signals [1,19].

#### 2.1.3. Osteoclasts

Osteoclasts are terminally differentiated myeloid cells that are located in shallow depressions on bone surfaces called Howship’s lacunae [13]. Osteoclasts are highly migratory, multinucleated, and polarized cells, which make them uniquely adapted to bone resorption [14].

A disruption in osteoclast formation and activity leads to bone disorders such as osteoporosis, where resorption exceeds formation and causes decreased bone density and increased bone fractures. Furthermore, osteoclasts have been associated with several other functions, including regulation of the hematopoietic stem cell niche through cytokine production [1].

#### 2.1.4. Bone-Lining Cells

Bone-lining cells are quiescent osteoblasts that are found lining the bone surface, being responsible for coupling bone resorption to bone formation. They anchor hematopoietic stem cells and provide them with appropriate signals to keep them in an undifferentiated state [1,20]. Thus, they play a crucial role in bone remodeling by gap junction communication with osteocytes in the bone matrix [10,20]. Specifically, these cells promote osteoclast differentiation and prevent undesired bone resorption by hampering direct interaction between osteoclasts and bone matrix [1]. Moreover, bone-lining cells prepare the bone surface by cleaning unmineralized collagen fibrils using matrix metalloproteinases (MMPs) and they deposit a smooth layer of collagen after remodeling [20].

#### 2.1.5. Bone Matrix

Bone matrix constitutes a complex and organized framework that plays a crucial role in mechanical strength and adhesive characteristics of the bone, consisting of an organic (30%) and an inorganic component (70%) [10,13]. The organic segment is mainly composed of collagenous proteins (90%) such as type I collagen, and 10% non-collagenous proteins (i.e., osteocalcin, osteonectin, osteopontin, fibronectin, and bone sialoprotein II), BMPs, and growth factors [1,10]. All these proteins have been shown to be crucial in relevant processes such as bone osteogenesis, mineralization, and remodeling [13].

The inorganic matrix consists predominantly of hexagonal hydroxyapatite (HA) crystals (Ca_10_(PO_4_)_6_(OH)_2_), making up 99% of the body’s storage of calcium, 85% of the phosphorous, and 40–60% of the magnesium and sodium [13]. In addition, ions such as bicarbonate, potassium, citrate, carbonate, fluorite, zinc, barium, and strontium are also present [1]. Biomineralization consist of deposition of HA crystals in the voids of a non-collagen protein and collagen-based scaffold, and allows interaction of organic and inorganic matrix [21]. Indeed, bone matrix proteins vary their concentration with age, nutrition habits, disorders, and treatments, which may increase bone fracture risk if disrupted [1].

### 2.2. Bone Remodeling Process

Bone remodeling constitutes a lifelong process that guarantees bone tissue integrity and mineral homeostasis by a coupling mechanism consisting of old or damaged bone removal (resorption) and new bone deposition (formation) [16,22].

The bone remodeling cycle comprises five different stages, depicted in Figure 2: activation, resorption, reversal, formation, and termination, occurring over several weeks (around 120 and 200 days in cortical and trabecular bone, respectively) [15,23].

#### 2.2.1. Activation

The bone remodeling process begins when bone in a quiescent state detects an initiating signal. This trigger can occur via different pathways, including mechanical forces, microscopic bone damage, and systemic hormones such as parathyroid hormone (PTH) [23,24]. The detection of the signal can result in osteocyte apoptosis [3,15], which involves the release of osteoclastogenic factors such as colony-stimulating factor 1 (CSF-1) and receptor activator of NF-κB ligand (RANKL) that favor osteoclast precursor recruitment and activation [1].

The bone-lining cells separate from underlying bone, exposing the bone surface and forming a canopy that can be resorbed. Thus, active BMU includes bone-resorbing osteoclasts which cover the newly exposed bone surface, preparing it for the deposition of replacement bone, and osteoblasts that secrete and deposit unmineralized bone osteoid [15].

#### 2.2.2. Resorption

Osteoclasts attach to bone matrix through Arg-Gly-Asp (RGD) binding sites and release enzymes and acids that help dissolve the mineral content of the organic matrix, forming an isolated environment known as the “sealing zone” and a ruffled border that provides an enhanced secretory surface area [10,16,23]. The remaining organic bone matrix is subsequently degraded by cathepsin K and matrix metalloproteinases [15]. The resorption phase lasts approximately 2–4 weeks and is concluded by mononuclear cells after osteoclasts undergo apoptosis [25], ensuring that excess resorption does not occur [23].

In addition, osteoblasts secrete cytokines and MMPs in order to degrade the unmineralized osteoid in bone surface and expose RGD adhesion sites to help osteoclast attachment [3]. Osteoclast differentiation, expansion, and survival depend on many factors and cytokines, such as CSF-1, RANKL, osteoprotegerin (OPG), interleukin (IL)-1, IL-6, PTH, and calcitonin [13,15].

#### 2.2.3. Reversal

Following the resorption phase, cells from an osteoblastic lineage, also called “reversal” cells, prepare the bone surface by removing unmineralized collagen matrix. Furthermore, they deposit a “cement line” of non-collagenous mineralized matrix to enhance osteoblastic adhesion [23]. The reversal cells are responsible for sending and receiving coupling signals, including bone matrix-derived factors such as transcription growth factor (TGF)-ꞵ, insulin-like growth factor, and BMPs [25,26]. Moreover, osteocytes up-regulate osteoblasts by the production of second messengers, such as nitric oxide and prostaglandins, which induce bone formation [15,16].

Osteoclasts are also thought to secrete cytokines such as IL-6 and to express a regulatory surface receptor called Ephrin receptor family [23]. Additionally, semaphorins comprise a large family of glycoproteins that are expressed by osteoclasts, and they have been proposed as inhibitors of bone formation during bone resorption [1]. The switch between resorption and formation has also been proposed to be triggered by a strain gradient within the lacunae, which may provoke sequential activation of osteoclasts (reduced strain) and osteoblasts (increased strain) [25]. Osteoclasts are then replaced by osteoblast-lineage cells, promoting the initiation of bone formation [15].

#### 2.2.4. Formation

Once early osteoblast progenitors have replaced osteoclasts in the resorption lacunae, they differentiate and start new bone formation [3]. Osteoblasts first synthesize and secrete a type 1 collagen-rich organic matrix, in order to fill in the cavities generated by osteoclasts [15,23]. Secondly, osteoblasts regulate osteoid mineralization, which involves the deposition of HA crystals amongst collagen fibrils. This process is mainly regulated by calcium and phosphate concentrations within membrane-bound matrix vesicles, and by local mineralization inhibitors such as pyrophosphate or proteoglycans [23,25]. Bone formation lasts approximately 4 to 6 months, completed by the apoptosis of the rest of the osteoblasts, which constitutes 50–70% of total osteoblasts [23].

#### 2.2.5. Termination

When the amount of bone formed equals the resorbed bone, the remodeling cycle terminates [15]. Osteocytes produce TGF-ꞵ and sclerostin, suppressing osteoclastogenesis and osteoblast differentiation through the Wnt pathway, respectively, which directly inhibits bone formation [17,23]. Osteoblasts form new bone until they revert to bone-lining cells that cover the bone surface [15,25]. Then, the resting bone surface environment is reestablished and preserved until the bone remodeling cycle is again initiated [3].

**Figure 2 materials-14-04896-f002:**
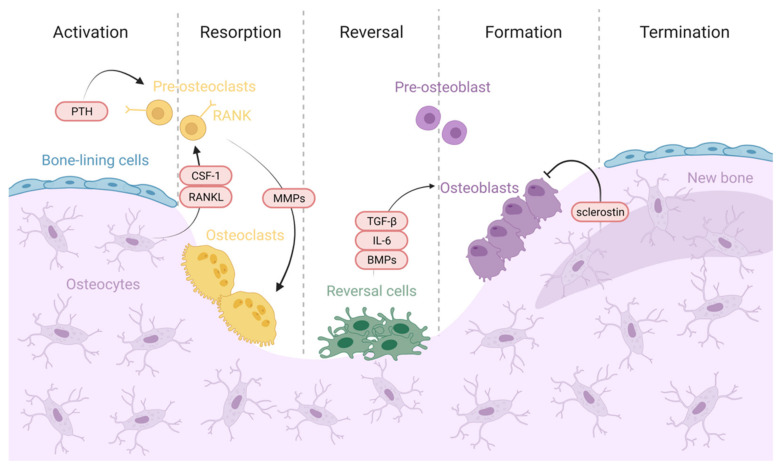
Bone remodeling process. Activation: Bone in quiescent state detects an initiating signal triggered by mechanical forces, microdamage, or systemic hormones such as PTH [23,24]. Osteocyte apoptosis is then induced [3,15], releasing osteoclastogenic factors such as colony-stimulating factor 1 (CSF-1) and receptor activator of NF-κB ligand (RANKL) that favor osteoclast precursor recruitment and activation [1]. Resorption: Osteoblasts secrete cytokines and matrix metalloproteinases (MMPs) in order to expose RGD adhesion sites to facilitate osteoclast attachment [3]. Osteoclasts dissolve mineralized matrix and organic bone matrix through hydrogen ions and MMP secretion, respectively [15]. Reversal: “reversal” cells remove unmineralized collagen matrix and deposit a “cement line” of non-collagenous mineralized matrix to enhance osteoblastic adhesion [23]. The reversal cells and the osteoclasts are responsible for sending and receiving coupling signals, including bone matrix-derived factors such as TGF-ꞵ, BMPs, and IL-6 [25,26]. Osteoclasts are then replaced by osteoblast-lineage cells, promoting the initiation of bone formation [15]. Formation: Osteoblast progenitors replace osteoclasts in the resorption lacunae, they differentiate and start new bone formation [3]. Osteoblasts continue to form new bone until they revert to bone-lining cells that cover the bone surface [15,25]. Termination: Osteocytes produce sclerostin, suppressing osteoblast differentiation and bone formation [17,23]. Then, the resting bone surface environment is reestablished and preserved until bone remodeling cycle is again initiated [3].

## 3. Bone Pathology

Bone disorders constitute the most common causes of severe long-term pain and physical disabilities [27]. They can result from several conditions, including tumors, genetic disorders, trauma, age-related disorders, and autoimmune diseases, among others [21,28]. In particular, disruptions in coupling bone resorption and formation during the bone remodeling cycle lead to loss of bone mass and skeletal integrity [29,30]. Among bone disorders, osteoporosis, Paget’s disease, osteoarthritis, fractures, and bone metastasis are the most frequent, and will be briefly described in this section.

### 3.1. Osteoporosis

Osteoporosis is the most common metabolic disease, characterized by low bone mass and structural deterioration of bone, causing bone fragility and an increased fracture risk [31]. Three different types of osteoporosis can be distinguished: primary, secondary, and idiopathic.

Primary osteoporosis includes senile and postmenopausal varieties, both related to natural gradual loss of bone mass. In postmenopause, estrogen is responsible for inhibiting bone resorption. Secondary osteoporosis occurs in patients with chronic diseases that provoke loss of bone mass, such as nutritional deficiencies, anorexia, alcoholism, and chronic liver disease. Idiopathic osteoporosis occurs in young adults or children with no endocrine abnormality or risk factor [27].

### 3.2. Paget’s Disease

Paget’s disease is a metabolic bone disease with controversial etiology. Genetics factors have shown to affect Paget’s development, specifically mutations that affect the RANK, OPG, and sequestosome 1. Paget’s patients show increased bone metabolic activity and imbalance between bone formation and resorption. Serum alkaline phosphatase constitutes a bone formation marker that allows diagnosis and monitoring of patients [27]. Bone pain, osteoarthrosis, deformities, fractures, and hypercalcemia are complications in Paget’s disease, among others [32].

### 3.3. Osteoarthritis

Osteoarthritis is the most common adult joint disease, appearing as a result from gradual loss of cartilage and development of bony spurs and synovial cysts at the ends of the joints [27]. It can be classified into primary osteoarthritis, which results from genetic causes and affects middle-aged people, and secondary osteoarthritis, related to metabolic diseases, endocrine disorders, etc. Risk factors for osteoarthritis include endogenous factors (age, obesity, sex, heredity, ethnic origin, and postmenopausal alterations), and exogenous factors (i.e., macrotrauma, prior joint surgery, and lifestyle factors) [27,33].

### 3.4. Autoinflammatory Diseases

Innate and acquired immunity disorders can cause autoinflammatory bone diseases by infiltrating innate cells into the bone with subsequent increased osteoclast activity, osteolysis, and bone remodeling [34]. Indeed, low-grade chronic inflammation leads to bone resorption in many rheumatic diseases, making up the link between the immune system and the bone. In autoimmune diseases, osteoclastogenesis might be triggered by autoreactive T cell-secreted IL-17 and RANKL. Moreover, Toll-like receptor signaling is thought to be involved in its modulation. The most common autoinflammatory bone diseases include chronic non-bacterial osteomyelitis; synovitis, acne, pustulosis, hyperostosis, and cherubism, among others [35].

### 3.5. Bone Metastasis

Bone metastasis constitutes a frequent consequence of many cancers and typically indicates a short-term prognosis in cancer patients [36]. Indeed, patients with bone metastasis have a higher mortality rate and suffer from pain and clinical complications such as fracture, spinal cord compression, and hypercalcemia, which seriously reduce the quality of life [37].

Bone metastases are classified as osteolytic, osteoblastic, or mixed, according to the interference with the bone remodeling cycle. Osteolytic metastasis is present in the great majority of bone cancers, destroying normal bone primarily mediated by osteoclasts, parathyroid hormone-related peptide (PTHrP) and RANKL. Osteoblastic metastasis is characterized by deposition of new bone, and occurs in prostate cancer and Hodgkin lymphoma, among others [36].

### 3.6. Therapeutics in Bone Disease

Broadly, most bone disease treatments only help to reduce the symptoms, but do not cause a total suppression of the disease. The mechanism of action for the majority of the drugs consists of disrupting the osteoclastic pathway and promoting the osteoblastic one [38]. Bisphosphonates constitute the most commonly used drugs in most bone treatments, mainly indicated for osteoporosis, Paget’s disease, and bone tumors [7]. These agents bind to HA and block the bone resorption mediated by osteoclasts. Although they increase bone mineral density (BMD) and reduce serum PTH, long-term treatment causes inhibition of both bone resorption and formation markers. Moreover, they cause gastrointestinal and cardiovascular side effects [7]. To date, the Food and Drug Administration (FDA) has approved four different bisphosphonate-based drugs for the treatment of osteoporosis, including alendronate, risedronate, ibandronate, and zoledronic acid [27].

Calcitonin is used in the treatment of Paget’s disease, binding to the calcitonin receptors located on the osteoclast surface. Calcitonin also presents analgesic properties that further help treat patients with bone pain [31]. However, it is only used in those patients with intolerance to bisphosphonate, since it shows a short duration of action, partial response, and acquired resistance. Denosumad, a monoclonal antibody, is an anti-resorptive pharmaceutical agent targeting RANKL signaling in osteoclast activation. It is indicated for anti-fracture in women with postmenopausal osteoporosis, and also in bone tumors [38]. It stimulates osteoclast differentiation and function and promotes osteoclast survival, increasing bone mineral density. In contrast, it causes hypercalcemia, nephrotoxicity, and seizures [7]. Raloxifene is approved for the treatment of osteoporosis and bone fracture. It is a selective estrogen receptor modulator that acts as an estrogen agonist on both bone and fat tissue, and as an estrogen antagonist on breast and uterine tissue. Raloxifene has been shown to cause cardiovascular events and venous thromboembolism [7].

Currently, bone metastasis can be treated by inhibiting cancer cell growth or by inhibiting the osteoclast activity. Chemotherapy is the most common approach to date to inhibit cancer cell growth, and efficient drug delivery to the bone metastasis is essential for success. Nevertheless, the skeleton presents significantly poorer blood supply than other tissues, decreasing the efficiency of drug delivery to bone and resulting in severe side effects caused by the lack of tissue specificity [37].

In the bone fracture framework, autologous bone grafting has been established as the gold standard technique for repairing bone tissue, consisting in bone harvested from the same individual receiving the treatment [21,28]. Autografts have been shown to be histocompatible and non-immunogenic, but they present significant limitations, such as donor site injury and morbidity, surgical risks (bleeding, inflammation, infection, and chronic pain), and resorption of the implanted bone [28]. Allografts provide an alternative method, particularly in areas with larger bone defects, based on transplanting donor bone tissue between genetically non-identical individuals of the same species [21,39]. However, they present reduced osteoinductive properties when compared to autografts [40], taking longer for a defect to be filled by native bone tissue [29], which leads to mechanical stress and immunogenesis [39].

Since no treatment option provides overall successful results in bone pathologies, recent research has been devoted to developing tissue engineering strategies that can overcome these issues. In this context, BTE has emerged as an excellent tool to design and implement new bone repair interventions. Fundamental notions behind this subject and the state of the art to date will be reviewed in the next section.

## 4. Scaffolds in Bone Tissue Engineering

BTE has arisen as a powerful tool for developing new bone treatment options to overcome shortages of existing bone graft materials and bone disease therapies. In recent decades, BTE strategies have been directed to produce bone constructs that mimic natural bone structure regarding mechanical strength and microstructure [41].

In this context, the design of accurate biomaterials for BTE should be broadly based on several components: (1) a biocompatible scaffold that resembles extracellular matrix (ECM), (2) osteogenic cells to build the bone tissue matrix, (3) morphogenic cues and growth factors to help guide osteogenic cells towards bone phenotype, and (4) material/host interactions mediated by immune cells that fulfill the bone healing process [39,40].

Ideally, the material should degrade following initial cellular colonization simultaneously to new bone formation. Then, the tissue will reestablish its function with no clinical intervention needed to remove the implant [42]. The key factors involved in the achievement of this smart biomaterial will be addressed in the following subsections, presenting the state of the art to date and the aspects to be considered in the material design.

### 4.1. Scaffold Composition

Scaffolds are temporary matrices built from biocompatible, bioactive, and biodegradable materials. An ideal scaffold should mimic the structure and function of the ECM of original bone, which will maintain a suitable environment and architecture for bone growth and development. Specifically, they should be based on a 3D structure that allows proliferation and homing, osteogenic differentiation, vascularization, host integration, and load bearing [6,43].

Since the different bone types present different remodeling period lengths, there is not one ideal scaffold material for all BTE purposes, being subject to the size, type, and location of the bone tissue to be regenerated [44]. First generation biomaterials include metals (mainly titanium derivates), synthetic polymers (such as Poly (methyl metacrylate) (PMMA) and Polyether Ether Ketone (PEEK)), and ceramics (such as alumina and zirconia), being bioinert and hardly interacting with host tissue. Second generation biomaterials stood out for their bioactive and biodegradable features, including synthetic and natural polymers, calcium compounds (phosphates, carbonates, and sulfates), and bioactive glasses. The third generation of biomaterials allow the addition of cells, biological factors, or external stimuli, among other instructive substances, to induce specific biological responses [45].

Most of the scaffolds that are currently used for BTE applications can be grouped into polymers, bioactive ceramics, composite materials, and nanomaterials.

#### 4.1.1. Polymeric Scaffolds

Generally, polymeric materials present reproducible and adjustable physiochemical characteristics such as pore size, porosity, solubility, biocompatibility, and immune response [46]. Both natural and synthetic polymers are valuable materials for BTE.

Natural polymers include proteins (such as collagen, gelatin, fibrinogen, elastin, keratin, and silk); polysaccharides (chitosan, alginate, hyaluronic acid, and cellulose); and polynucleotides (DNA, RNA) [47,48]. Extracellular matrix ECM-based scaffolds have been also proposed due to their similarity to bone tissue. All of them have shown great biocompatibility, osteoconductivity, controlled biodegradation, and low immunogenicity [6,47]. However, they present poor mechanical properties, uncontrollable degradation rate, and reduced tunability [6,29].

Synthetic polymers consist of aliphatic polyesters such as poly(lactic-acid)(PLA), poly(glycolic-acid)(PGA), poly(caprolactone) (PCL), polypropylene fumarate (PPF), and PEEK and their copolymers [47]. They are biocompatible, biodegradable, and display a controlled degradation rate. Moreover, synthetic polymers offer tunable properties and structure, allowing a controllable degradation rate and mechanical properties [29]. These features guarantee predictable, reproducible, and cost-effective production of BTE scaffolds. However, they have shown reduced bioactivity and osteoconductivity when compared to natural polymers [48].

Hydrogels are hydrophilic polymeric 3D networks that constitute some of the most promising biomaterials in BTE applications, due to their flexibility and bioactivity. In contrast to rigid scaffolds, hydrogels can actively interact with the host tissue, allowing controlled release of cells and growth factors. Unfortunately, because of their low stiffness, they cannot be utilized in the repair of load-bearing lesions, such as large fractures of long bones [49].

#### 4.1.2. Bioceramics

Bioceramics are inorganic biomaterials that almost mimic bone tissue composition, providing osteoconductive properties, nontoxic degradability, high compressive strength, and long shelf-life [38].

Most common bioceramics include coralline, HA, tricalcium phosphate (TCP), and calcium silicates [6]. Particularly, calcium phosphate ceramics (CPCs) have been extensively studied for bone tissue repair as tunable bioactive materials, providing the pores for tissue ingrowth and viability [50]. The capacity of these materials to elicit osteogenesis and angiogenesis effects through releasing biologically active ions has attracted great interest [46].

Bioactive glasses are usually composed of calcium-containing silicates. For instance, 45S5 Bioglass contains 45wt% SiO_2_, 24.5wt% CaO, 24.5wt% Na_2_O, and 6.0wt% P_2_O_5_, enabling an increased regeneration rate when compared with HA-based ceramics. Bioactive glasses are thought to create a bone-like apatite layer on the material surface, releasing ions such as Ca^2+^, PO_4_^3−^, and Si^4+^ and hence promoting osteogenesis. Moreover, mesoporous bioactive glasses have emerged as an excellent tool for BTE, since they can be easily loaded with drugs or biomolecules, due to their large pore surface [46].

However, bioceramics are inherently brittle, making them difficult to handle in implantation. To overcome this issue, several strategies have been explored. For instance, coating bioceramics with polymers such as Poly Lactic-co-Glycolic Acid (PLGA), chitosan, or Polyethylene Glycol (PEG) has been shown to improve scaffold mechanical properties and also osteogenic capacity [50]. Furthermore, the composition of bioceramics can be altered to modulate the fracture toughness. In particular, doping β-TCP scaffolds with SiO2 (0.5%) and ZnO (0.25%) raised the compressive strength 2.5-fold and increased cell viability up to 92% [47].

#### 4.1.3. Composite Materials

Composites combine two or more materials with different properties that may improve scaffold features when compared to their individual components, such as processability, printing performance, mechanical properties, or bioactivity [46]. They can be formed from co-polymers, polymer–polymer blends, or polymer–ceramic composites [6].

Composites can be based on co-polymers derived from two or more monomers, such as PLGA, which combines poly lactide and polyglycolide and shows adequate biodegradability and ease of fabrication. In this context, hydrogels seem to mimic ECM topography and hence they are ideal for cell adherence, proliferation, and differentiation. Hydrogels can be both natural (agarose, alginate, and gelatins) and synthetic (poly(vinyl alcohol) based) [6].

Polymer–ceramic composites resemble the natural bone, which is also a composite material made of inorganic HA crystals and collagen fibers. As has been previously mentioned, calcium phosphate (CP)–polymer composites show mechanical integrity and bioactivity, being successfully applied in bone regeneration. Moreover, the incorporation of bioceramic and bioglass particles, carbon nanotubes, or magnesium-based particles seems to improve scaffold mechanical properties [6].

#### 4.1.4. Nanomaterials

In recent decades, nanosized materials have shown better bioactive properties than micro-sized ones, attracting much interest in BTE. In particular, nanoceramics appear to be promising biomimetic candidates, since bone can be considered a nanocomposite [6]. Specifically, the collagen fibers and HA crystals are nanosized in diameter, as well as all living molecular building blocks. Thus, bone cells are predisposed to interact at the nanoscale, enhancing adhesion, proliferation, and differentiation processes [40]. Indeed, changes at the nanoscale level of tissue hierarchy may have further effects on migration and cell signaling [42,51].

Furthermore, nanoparticles (NPs) can be incorporated into scaffolds, leading to superior mechanical properties, increased surface area, and roughness, as well as osteointegration, osteoconduction, and osteoinduction [40]. Consequently, biomaterial design has focused on the introduction of nanoparticles to elicit directed cellular behavior while imparting structural and mechanical advantages. Moreover, NPs can be applied as drug delivery and cell tracking systems, due to their functionalization versatility, allowing a significant improvement in prospective therapies.

For instance, HA–collagen nanocomposite systems, for example, are emerging rapidly and showing promise [52]. Yan et al. reported that the incorporation of calcium phosphate NPs into silk fibroin (SF)-based scaffolds induced increased bone formation after 3 weeks of implantation in rat femur, compared to bare SF scaffolds [53]. Wang et al. presented magnetic lanthanum (La)-doped HA NP/chitosan scaffolds (MLaHA/CS) that were able to recruit rat bone marrow MSCs and modulate host-to-scaffold immune responses via macrophage polarization [54].

In addition, multi-walled carbon nanotubes (MWCNTs) have been shown to improve the compressive strength and elastic modulus of 45S5 Bioglass^®^ (Gainesville, FL, USA) scaffolds [55]. Mesoporous silica nanoparticles (MSNs) present high bioactivity and excellent biocompatibility, being also particularly suitable for loading bioactive agents due to their highly controllable morphology and structure [56]. Although nanosystems have not yet achieved clinical implementation, in vitro and in vivo studies have been performed in recent years, showing the great potential of this technology [9].

### 4.2. Scaffold Properties for BTE

The modulation of the scaffold characteristics is key in the final success of the biomaterial for clinical implementation (Table 1). In general terms, a scaffold must be biocompatible, not elicit an immune response, and be biodegradable into non-toxic components with a controllable degradation rate to match new bone formation [21,44]. Particularly, bioincompatibility could lead to immune system activation, tumor formation, and inflammation processes [6,40]. Recent research has been directed to immune-inert and immunomodulatory biomaterials that are potentially able to regulate host immune response [57].

Biodegradability of the material occurs through chemical or enzymatic breakdown of the material over time when introduced into living organisms [21]. Degradation products should be non-toxic components and they must be either recycled or excreted with negligible interference with other organs [6].

Scaffold bioactivity constitutes another key feature, comprising osteoconductivity and osteoinductivity processes [54]. Osteoconductive scaffolds facilitate bone deposition on the material surface by interacting with the surrounding living tissues. Bioactive scaffolds are hence designed to enhance cell migration, tissue formation, and integration, overcoming conventional passive biomaterial limitations [6]. Moreover, osteoinduction properties provide the scaffolds with the ability of promoting the differentiation of progenitor cells down an osteoblastic lineage [10,52].

Structurally, a scaffold must be a 3D network with highly interconnected pores that allow cell growth and mass transfer of nutrients and metabolic waste [44]. Moreover, a large surface area becomes crucial for vascularization and tissue infiltration [54]. Broadly, macroporosity might promote osteogenesis by facilitating cell and ion transport, while microporosity improves surface area for protein adsorption, increasing ionic solubility in the microenvironment, and providing attachment points for osteoblasts [58].

Typically, pores should be bigger than 300 µm to facilitate new bone formation and vascularization, since they provide enough space for oxygen and nutrient supply [54]. BTE scaffolds broadly range from 50 to 900 µm. Small pore sizes (75–100 µm) have shown successful bone regeneration in vitro and are thought to promote angiogenesis. On the other hand, macropores might promote osteoinductivity by mediating vascularization. A pore size range from 200 to 500 μm has shown the optimal tissue penetration vascularization in vivo [6,21].

Pore interconnectivity has been shown to positively influence bone deposition rate and depth of infiltration both in vitro and in vivo [58]. Percolation diameter is a key parameter in scaffold architecture, which defines the diameter of the largest tracer that can cross the interconnected pores of a scaffold. Thus, it determines the scaffold interconnectivity and hence the size of cells and nutrients that can pass through them. The bottleneck dimension describes the diameter of connections between pores, with it being established that the 700–1200 µm range is the optimal dimension for bone substitutes [54,59].

Other relevant surface features are surface roughness, surface free energy, surface charge, and surface topography, which can be easily tailored by chemical treatment or incorporation with bioactive molecules or artificial ECM [60]. Apparently, the biomaterial surface is the most critical aspect in the host immune response upon implantation and becomes responsible for avoiding macrophage adhesion and activation, as well as their fusion into foreign body giant cells [6]. Specifically, roughened surfaces have shown increased osteointegration by favoring epithelial attachment, when compared to smooth surfaces, improving implant durability and function [61].

Likewise, the scaffold is temporarily responsible for mechanical support and stability at the tissue formation site. Thus, mechanical properties are also considered crucial in BTE scaffolds and should be adapted to the site of implantation to minimize risk of stress shielding, implant-related osteopenia, and subsequent refracture [6]. In general terms, the scaffold material should be sufficiently robust to host cells and resist contraction forces during tissue healing [44].

Properties such as elastic modulus (a measure of the stiffness of a solid material), tensile strength (capacity of a scaffold to cope with loads tending to reduce size), and fatigue strength (the highest stress that a material can hold for a given number of cycles without breaking) should be similar to those of natural bone in order to ensure bone mechanical strength [62]. However, bone presents diverse and dynamic mechanical properties, tightly related to its complex hierarchical structure [63]. Specifically, the elastic moduli of human bone tissue usually varies between 1 and 20 GPa (around 2.0 GPa and 14-18 GPa for trabecular and cortical bone, respectively) [54,64], and the tensile strength of cortical and cancellous bones is 50–150 MPa and 10–100 MPa, respectively [65].

**Table 1 materials-14-04896-t001:** Scaffold properties for design considerations.

Surface area	Crucial for cell–scaffold interactions, facilitating vascularization and tissue infiltration	[54]
	Macroporosity might promote osteogenesis by facilitating cell and ion transport	[58]
	Microporosity improves surface area for protein adsorption, increasing ionic solubility and attachment points for osteoblasts	[58]
Pore size	Pores > 300 µm facilitate new bone formation and vascularization	[54]
	75–100 µm pore size is thought to promote angiogenesis	[6]
	Pore size range from 200 to 500 μm results in optimal tissue penetration vascularization in vivo	[6,21]
Pore interconnectivity	Enhanced bone deposition rate and depth of infiltration	[58]
	Optimal diameter of connections between pores ranges from 700–1200 µm	[54,59]
Surface topology	Roughened surfaces promote osteointegration and favor epithelial attachment	[61]
Mechanical properties	Young’s modulus should be close to 7–30 GPa and a tensile strength of 50–151 MPa	[62]
	Compressive strength should be comparable to cortical bone (100–230 MPa)	[62]
	Degradation rate should match the growth of native ECM to ensure scaffold mechanical support	[21,44]

Likewise, the mechanical strength can be weaken by high porosity and degradation rates [21]. Although higher porosity and pore size promote bone regeneration, both can impair the mechanical properties of the scaffold [54]. In addition, the degradation rate should match the growth of native ECM to ensure mechanical support throughout the lifecycle of the scaffold. This is also influenced by the porosity and pore size: if a scaffold degrades too quickly, it may lead to mechanical failure. In the same way, if the degradation rate is not sufficiently quick, an inflammatory process is triggered towards the foreign material, impairing tissue regeneration [62].

Therefore, a trade-off between biomechanical properties and degradation kinetics becomes necessary. Ideally, a scaffold material should provide sufficient mechanical stability while the newly formed bone tissue substitutes the scaffold matrix [44]. To achieve that, overall porosity can be tailored to adjust site-specific biomechanical requirements. Several studies claim that changes in macroporosity, and more specifically minimizing closed porosity, have shown a more significant effect in mechanical properties [58,66].

Considering all these scaffold requirements, the ideal scaffold is thought to have a compressive strength comparable to cortical bone (100–230 MPa), with a Young’s modulus close to 7–30 GPa and a tensile strength of 50–151 MPa [62]. Scaffolds fabricated from biomaterials with a high degradation rate should avoid high porosities, whereas slowly degrading biomaterials can be highly porous and still present adequate mechanical integrity [44].

Furthermore, stimulus-responsive scaffolds have attracted much interest in terms of modulating their physicochemical and mechanical properties, as well as the degradation profile in response to stimuli (including photoirradiation, mechanical, magnetic, and electrical forces, and changes in pH or enzymatic reactions) [42]. For instance, thermoresponsive biomaterials based on different polymers are tailored to have transition temperatures to gelate when the temperature is elevated to physiologic temperature [67,68,69]. Recent studies have designed piezoelectric systems that might accelerate new tissue formation in response to mechanical loading. In addition, NP-reinforced polymers can offer better mechanical, electrical, and thermal properties [42]. The development of clear strategies for optimizing and balancing all these properties becomes crucial for BTE translation in clinics.

### 4.3. Scaffolds as Vehicles of Cells and Growth Factors

Although cell-free biomaterials have been used to stimulate bone regeneration, exogenous cells with osteoblast differentiation potential must be provided when osteoprogenitors are not available in the damaged tissue [49]. Thus, scaffolds can act as vehicles for different types of cells in order to promote bone formation in vivo by differentiating towards the osteogenic lineage [6]. These cells may be expanded ex vivo before the implant, and the most commonly utilized are embryonic stem cells, induced pluripotent stem cells, MSCs, and genetically modified cells [6].

Embryonic stem cells (ESCs) can differentiate into a number of specialized cells present in the bone, including MSCs, osteoblasts, chondrocytes, endothelial cells, etc. [70]. Moreover, ESCs show rapid proliferation and self-renewal over long periods of time, making them ideal tools in BTE. For instance, Tang et al. [71] described in 2012 how hESCd-MSC-encapsulating hydrogel microbeads in a macroporous CPC construct undergo osteogenic differentiation and bone mineral synthesis. However, ESCs have resulted in ethical concerns in various countries, as well as immunogenicity and tumorigenicity, which strongly hamper their clinical application [72].

In this context, induced pluripotent stem cells (iPSCs) show analogous advantages to ESCs, and they avoid the risk of immune rejection, since they derive from patient somatic cells. In many studies, iPSCs and MSC-like derived iPSCs cultured on an HA/TCP scaffold showed osteogenic differentiation both in vitro and in vivo [73]. However, incomplete or false reprogramming may result in teratomas, especially in the presence of oncogenes or reprogramming viral methods. Prior in vitro differentiation into MSCs and direct differentiation into osteoblasts, among other strategies, have been reported to avoid tumorigenicity associated with pluripotency [72].

MSCs are relevant in osteoblast differentiation and immunomodulatory capability, being critical for stimulating tissue regeneration. In particular, MSCs secrete immunosuppressive and anti-inflammatory cytokines, regulate T cells, and promote a local healing response by stimulating proliferation and differentiation of host stem cells [49]. In MSC-based tissue engineering approaches, MSCs are incorporated in scaffolds that facilitate the local delivery of MSCs into the bone defects. Scaffold colonization can occur either by cell culture prior to scaffold creation, or by simultaneous deposition of cells and biomaterials [74].

Transplanted MSCs have been shown to proliferate and differentiate into an osteogenic phenotype [75]. However, their short lifespan after implantation becomes a major limitation, mainly due to poor adherence to the matrix, ischemia, and low glucose levels. Cells colonize the defect, proliferate, and differentiate into mature osteoblasts, promoting the formation of new bone tissue. However, the lack of a vascular network upon implantation impedes cells survival in the site of implantation. In this framework, different strategies can be followed to promote cell and rapid vascularization in the defect area [49]. For instance, the association of oxygen carriers with BTE scaffolds has been shown to improve survival and the bone regeneration capacity of the cells [49]. Moreover, several studies have sought to mimic the initial stages of endochondral ossification by chondrogenic priming of MSCs, which can significantly increase osteogenic differentiation and mineralization, even to a higher extent than culturing cells in osteogenic growth factors alone [76].

Thus far, bone healing has been shown to be improved when utilizing MSCs with metals, ceramics, and polymeric and composite materials [74]. The scaffolds should provide the proper microenvironment for cells to survive, proliferate, and differentiate in vitro and in vivo. For instance, a system comprising PLGA membranes incorporating BMP-2-loaded microspheres and rat mesenchymal stem cells (rMSCs) with their Smad ubiquitin regulatory factor-1 (Smurf1) expression knocked down was evaluated in a rat calvaria, a critical size defect by Rodríguez-Évora et al. [77], showing a defect repair of 85% after 8 weeks.

The interdependence between macrophages and osteoblasts is also essential for the resolution of the inflammatory process, since it has been demonstrated that some macrophages are involved in particle clean up and the up-regulation of osteolytic and pro-inflammatory cytokines, including IL-6 and tumoral necrosis factor (TNF)-α [42]. Further research has shown that the use of macrophage-recruiting agents combined with platelet-rich plasma in a hydrogel induced bone formation [78].

Still, bone tissue biology depends on the action of growth factors (GFs), which control cell behavior through specific receptors on the target cell membrane. In a bone regeneration context, these factors diffuse signals at the defect site through the ECM, facilitating the migration of progenitors and inflammatory cells to initiate the healing process [79].

Therefore, incorporation of GFs into the scaffold biomaterial is thought to improve osteogenesis and angiogenesis, limiting excessive bone formation and accelerating the healing process [47,79]. Thus, bone tissue engineering strategies pursue the combination of cells and engineering materials with growth factors. The main factors involved in bone healing have been mentioned in previous sections and can be grouped into several families: osteogenic (BMPs, activins, among others), angiogenic (such as platelet-derived growth factor (PDGF) and vascular endothelial growth factor (VEGF)), inflammation control (e.g., TNF-α, interleukins, interferon-γ), and systemic factors (vitamin D, growth hormone, calcitonin, PTH) [47,80].

Among them, the use of BMP-2 and BMP-7 has been incorporated in FDA-approved devices for bone regeneration, showing a clinical performance comparable to autografts in certain BTE applications [79]. Still, large doses of recombinant BMP-2 are required to promote bone formation, sometimes leading to adverse effects (e.g., heterotopic ossification and osteolysis). In this context, BMP-9 has emerged as a potential alternative, described as the most potent BMP in osteogenic differentiation. Indeed, it has been shown to completely close rat critical-sized calvaria defects when loaded into a microparticle (MPs)–hydrogel scaffold [81]. Furthermore, BMP-2 treatment has been shown to fail to regenerate articular cartilage structures such as joints, where BMP-9 has been proven to stimulate regeneration. For this reason, the sequential administration of BMP-2 and BMP-9 has resulted in regeneration of both bone and joints, providing the adequate environment for bone regeneration [82].

In addition, molecules such as N-acetyl cysteine have been exploited to promote runt-related transcription factor 2 (RUNX2) expression, which is involved in osteogenic differentiation of MSCs [80]. A product based on β-tricalcium phosphate carrying rhPDGF (Augment^®^ Bone Graft, BioMimetic Therapeutics, (Franklin, TN, USA)) is also commercially available to treat foot, ankle, and distal radius fractures in the USA [80]. Moreover, local delivery of angiogenic growth factors such as VEGF and basic fibroblast growth factor (bFGF) has been shown to accelerate vascularization through recruitment of endothelial progenitor cells (EPCs) [40].

Despite the success of numerous bone-promoting growth factor-based products, several complications have been reported related to their clinical utilization. Specifically, inflammation, pleiotropic effects, hypotension, and teratogenic effects constitute some of the main concerns. In this regard, carriers with reliable release kinetics could help in the adoption of this therapeutical strategy [80]. Additionally, proper dosage of GFs has been shown to affect the quality of the vessels. For this reason, the addition of growth factors that stimulate the recruitment of smooth muscle cells or pericytes has been considered to help vessels stabilize and mature [40].

Likewise, scaffolds loaded with soluble molecules such as antibiotics and chemotherapeutic agents that can be delivered in the environment have also been exploited. An extensive review on the clinical application of bone-targeted drug delivery for bone malignancies is presented by Ferracini [83], showing a promising route to treat osteomyelitis, bone metastasis, OS, osteoarthritis, osteonecrosis, and delayed/non-unions, among other bone disorders.

## 5. Cell–Biomaterial Interactions beyond Microenvironment

Despite the recent advances in BTE, utilization of biomaterials has barely translated to human clinical trials, mainly due to a lack of studies about the interaction between biomaterials and host cells. Particularly, cell–material interactions occur at micro- and nanoscales, where cells physically contact the ECM, altering their response. As has been mentioned in Section 4.2, tuning 3D scaffold chemistry, topography, and stiffness can help selectively guide cell proliferation and differentiation within the bulk material. A better understanding of the effect that scaffold parameters exert on host cells might help map out the future path for BTE translation.

### 5.1. Cell Response to Biomaterial Chemistry

Biomaterial surface chemistry is somehow the most manageable parameter to control its effect on cell behavior. At the macroscale, coating approaches seek to increase cell adhesion or specifically select cell types for cell attachment.

There are a variety of inorganic and organic compounds, such as collagen, fibronectin, gelatin, or poly-L-Lysine, that have been used for covering substrates. Interestingly, culture surfaces have been frequently ornamented with adhesion-derived peptides, such as RGD at 1 pmol/mm^2^ [84], that enhance cell attachment in 2D. To avoid degradation or detachment issues of the coated compound, normally, the substrate chemistry is modified, or ligands are added as presenting proteins. Biomaterials functionalized with RGD peptides have been shown to direct cell adhesion, migration, and differentiation in numerous tissues, including bone, neural, endothelial, liver, and cancer cells [85]. It has been found that RGD density positively regulates osteoblast mineralization [86]. RGD sequence interaction with integrins is further modulated by RGD conformation (cyclic penta- and hexapeptide), which leads to specific integrin affinity. In the case of osteogenic cells, functionalization of PMMA and PEG-PLA substrates with RGD favors osteoblast adhesion, proliferation, and bone formation through selective engagement of α_v_β_3_ and α_v_β_5_ integrins [87,88]. Moreover, Gly-Phe-Hyp-Gly-Glu-Arg (GFOGER)-functionalized hydrogel leads to superior osteogenic differentiation and bone formation of human mesenchymal stromal cells, compared to RGD-functionalized hydrogel both in vitro and in vivo [89].

Patterning at the microscale is a potent tool to control the cell shape, location, and density. Combining non-adhesive regions (normally treated with PEG) with adhesive regions (a variety of cell adhesion proteins (CAPs)) can generate a multitude of patterns and cell spots, which can be especially useful in printing methods [90,91].

Using fibronectin micropatterned material, McBeatch et al. [92] demonstrated that cell shape regulated human mesenchymal stem cell (hMSC) differentiation. Thus, hMSCs allowed to adhere, spread, and flatten were differentiated into osteoblasts, whereas those that did not attach formed spherical cells that resulted in adipocytes. Furthermore, modeling techniques can be used in co-culture assays where two or more types of cells are selectively adhered by controlling their cell–cell contact, spacing, and microenvironment interactions across flow-through microfluidic channels [93,94], by stamping with a stencil-based approach [95], or by seeding on surfaces that dynamically switch from cell adhesive to cell repulsive [96,97]. Advances in high-throughput chemical interaction approaches permit a large screening of cell–ECM, cell–biomaterial, or cell–cell interaction at the nanoliter scale. Recently, microarrays of biomaterials have been generated by robotic spotters that dispense and immobilize nanoliters of materials to test the interaction of stem cells with extracellular proteins [98].

### 5.2. Cell Response to Biomaterial Topography

Cell behavior can be regulated by topographical signals from the underlying substrate. In this regard, micro- and nanofabrication methods that reproduce in the substrate the topographical cues of the cell niche have become crucial [99].

Topography represents the different structures that exist on the surface of biomaterials. The majority of biological surfaces contain non-organized features at the macro- and microscale, being more organized at the nanoscale [100]. Cells have been shown to respond to topographical cues down to 5 nm, affecting proliferation, gene expression [101], cell adhesion [102], motility [103], alignment, differentiation [104], and matrix production [105].

At the microscale, cell spreading directly correlates to the substrate exposure to ECM proteins, being topography independent. Osteoblasts [102] and endothelial cells [106] have demonstrated a reproducible response to surface texture. Polarization or cell alignment generally rises with increasing groove depth and decreasing groove spacing [106], although the response to substrate topography is highly dependent on cell type. At the nanoscale, several reports have suggested that cells can sense and respond to features of 10–30 nm in size [107]. Indeed, the repetition of similar patterns has the greatest effect and provides the most predictable results [108,109]. Proteomic-derived studies have shown that human osteoprogenitor cells respond to nanosized pits and pores in asymmetrical patterns, favoring differentiation and matrix production [110].

Altogether, advances in domineering topography would provide an alternative to exogenous biological stimulation, diminishing fabrication costs and avoiding its adverse effects. For instance, intra-articular injections of TGF-ꞵ result in osteophyte formation in the murine joint [111].

### 5.3. Cell Response to Biomaterial Elasticity

Cells are also affected by physical and mechanical properties (stiffness or rigidity) of the substrate. Stiffness constitutes the resistance of a solid material to deformation, being defined by elastic modulus (E) and reported in “Pascals” (Pa). Bones usually range from 10–20 × 10^9^ Pa, even though most of the tissues have elastic moduli in the range of 10^1^–10^6^ Pa, lower than most culture surfaces, such as TCP (3–3.5 × 10^9^ Pa).

The rigidity of a material hence regulates cell adhesion, spreading, and migration. In particular, the mechanotaxis phenomenon induces cells to migrate preferentially towards stiff surfaces in 2D cultures [112]. Moreover, elasticity has also been proven to trigger the differentiation of MSCs. For instance, Engler et al. [113] demonstrated that MSCs cultured on collagen-coated material with stiffness variability can differentiate into neuronal, myogenic, and osteogenic lineages.

### 5.4. Cell Response to Mechanical Deformation

Elongation or compression of a material (2D) or matrix (3D) can lead to cell alignment in the direction of the applied load, resulting in elongated morphology in both 2D and 3D cultures [114,115]. In vitro, cyclic strain in both tensile and compressive loading enhances osteoblast mineralization [116]. In this context, bioreactors capable of applying physiologic loads have been fabricated to selectively produce and align extracellular matrix in a number of tissues, including bone [117], cartilage [118], and tendon [119].

### 5.5. Organ-on-a-Chip 3D Culture

Organ-on-a-chip models have arisen as a powerful means to mimic the 3D in vivo environment, since 2D culture lacks many native tissue aspects (such as microfluidic flow) [120]. An organ-on-a-chip is composed of a 3D culture system with dynamic flow through microchannels, achieved by integration of knowledge on microfluidics, microfabrication, and bioreactors. Cells can be exposed to flow pressure, topographical signals, and stimulation by embedded biomolecules and microelectrodes. The result can then be registered using microsensors [121] which will detect and control changes on a continuous basis, allowing real-time monitoring [122].

### 5.6. Extracellular Matrix and Cell–Biomaterial Interactions

ECMs are composed of noncellular biological materials, including (1) fibrillar, structural, and adhesive proteins such as collagen, elastin, and fibronectin; (2) amorphous matrix macromolecules like proteoglycans, glycosaminoglycans, and hyaluronan; and (3) growth factors, cytokines, and hormones. Biophysical properties of the ECM include strain, viscoelasticity, topography, and permeability [123].

At the tissue level, the ECM acts as a scaffold that provides structural and biomechanical support. At the cellular level, the ECM acts as a biophysical medium in which cells can attach and function. Moreover, the ECM also constitutes a reservoir for bioactive molecules such as growth factors and cytokines [123]. Cells interact with the ECM mainly through transmembrane adhesion proteins that are primarily integrins. Integrin–ECM ligand interactions are regulated by extracellular divalent cations (Ca^2+^ and/or Mg^2+^) and by the conformation of extracellular portion of integrins [124].

To stimulate cell–biomaterial interaction, ECM molecules and mimetic adhesive peptides can be introduced into scaffolds to reproduce biophysical and biochemical tissue properties, hence allowing the targeting of specific integrins and/or other receptors. In bone, fibronectin-specific α_5_β_1_ and collagen-specific α_2_ꞵ_1_ integrins are critical in bone marrow MSCs for osteoblastic differentiation [125,126].

### 5.7. Effect of Mechanical Forces on Cells and Tissues

Julius Wolff detailed in 1892 how structural bone remodeling was influenced by mechanical forces [127]. The evidence that mechanical forces rule tissue remodeling was one of the first insights into mechanosensing in biological systems and later became known as Wolff’s law. Broadly, cellular mechanosensing can be classified into: (1) focal adhesion and mechanosensing at the ECM, (2) cytoskeletal mechanotransduction, and (3) nuclear mechanotransduction (Figure 3).

Focal adhesion consists of multiprotein frameworks composed of integrins that create a mechanical link between the cell’s cytoskeleton and the local microenvironment [128,129]. The canonical model of cell adhesion is based on integrin activation, which leads to Rho and Rac protein activation [130] and subsequent recruitment of focal adhesion-associated proteins (e.g., vinculin, talin, and paxillin) [131]. These associated proteins sense mechanical tension across the focal adhesions [132]. In this sense, TGF-α mechanosignaling has been shown to be involved in the connection between endothelial cells and cancer [133].

Cytoskeleton mechanotransduction is based on focal adhesion complex activation, which triggers actin filaments and non-muscle myosine II to polymerize and rearrange into stress fibers [134]. Forces applied at focal adhesions can be passed on to the nucleus either through signal molecules or direct mechanical forces (Figure 3) [135]. Nuclear envelope spectrin repeat proteins (nesprins) are located on the outside of the nuclear envelope, allowing attachment of actin filaments and microtubules to the nucleus [136]. The generated structures are also known as “linker of nucleoskeleton and cytoskeleton” (LINC), and allow direct transmission of mechanical forces to the chromatin [137]. Major mediators of mechanosignaling are the transcriptional regulators YAP and TAZ (YAP/TAZ), which are crucial in the Hippo signaling pathway [138]. The tension on LINC proteins causes nuclear deformation, which favors YAP/TAZ nuclear entry and subsequent transcription of genes related to proliferation and migration [139,140].

### 5.8. Mechanical Forces in the Bone

Osteocytes are considered mechanosensors, since they respond to mechanical forces (principally to shear stress [141]), releasing prostaglandins and transmitting signals to other bone cells and the matrix through channels or cell–cell gap junctions, mainly formed by connexin43 [142]. Bone loss caused by a sedentary lifestyle, limb paralysis, or microgravity during space flight has proven the positive effects of physiological loading on skeletal tissues [143].

The compact bone matrix withstands compressive forces that range from 0.04–0.3%, limiting the transfer of force to the cells. Indeed, in vitro studies have concluded that strains must be 1–10% to induce cellular responses. Unmineralized bone matrix around osteocytes is more permeable than mineralized bone, creating lacuna–canalicular porosities for interstitial fluid flow that is mainly sensed by the osteoclasts [143]. Exercise-stimulated bone remodeling may be the response of osteoblasts to interstitial fluid flow [144].

In this sense, a recent elegant work showed that running exercise, in a murine model, control osteolectin^+^ cell content. Osteolectin-positive cells include osteoblasts, a subset of stromal cell leptin receptor-positive cells (LeptinR^+^), osteocytes, and hypertrophic chondrocytes in the bone marrow. Consequently, an increase in osteolectin levels activates osteogenesis. Osteolectin binds to α11ꞵ1 integrin, triggering Wnt pathway activation, which is necessary for the osteogenic response, revealing a new mechanism for maintenance of adult bone mass [145,146].

## 6. Computational Modeling

### 6.1. Bone Mechanobiology

Computational modeling has emerged as an excellent means to improve, predict, and optimize the clinical potential in terms of tissue-level phenomena, such as cell proliferation and differentiation, tissue growth, adaptation, and maintenance [12]. Cell incorporation into artificially engineered matrices has been commonly used in tissue engineering. As has been mentioned, most of the dynamic cell processes, such as growth, differentiation, or migration, are somehow controlled by mechanotransduction [147].

Although it is possible to mechanically stimulate bone and quantify changes at tissue levels in experimental techniques, it is extremely difficult to simultaneously describe and predict cellular and molecular mechanisms that underlie those changes. For this reason, computational mechanobiology has focused on elucidating the cell stimulation and on modeling individual bone cells to predict tissue differentiation. To do so, mechanoregulation algorithms and cell models for predicting mechanobiological changes have been developed [148].

Among mechanoregulation algorithms, two different approaches can be distinguished. First, algorithms based on the fact that tissue differentiation can be regulated by mechanical stimulation. For instance, Lacroix et al. proposed a poroelastic model based on the finite element (FE) method that was able to simulate direct periosteal bone formation, endochondral ossification in the external callus, stabilization during bridging, and resorption. The model successfully predicted tissue differentiation in a rabbit bone chamber and during osteochondral defect healing [148].

The second approach considers that musculoskeletal tissues also adapt to mechanical loading through changes in mass and shape. Studies based on tissue growth were initiated by Heegaard et al. [149], who reported a model to describe how the stresses generated by joint motion may affect the growth, leading to a congruent articular surface. Particularly, they developed a planar biomechanical model of the proximal interphalangeal joint and FE analysis to simulate joint kinematics and stress distribution resulting from muscle contraction [148].

Continuum methods such as FE simulations allow for examining mechanical experiments on cells and in vivo mechanical environments. These models include single cell–biomaterial interaction models; cell population–biomaterial interaction models; and indirect mechanistic models in tissue engineering [147]. Among them, Byrne et al. [150] and Sanz-Herrera et al. [151] simulated how scaffold design parameters such as porosity, Young’s modulus, and dissolution rate influence tissue growth.

### 6.2. Cell Adhesion

Numerous models for the analysis of adhesion have been developed to date. Among them, molecular models take place at the protein level, focusing on the ligand linkage between a cell and the ECM. In these models, steered molecular dynamics (SMD) allow computational modeling of molecular structure unbinding by applying an external force. Thus far, SMD has provided some insights on fibronectin-integrin binding and also on biotin–streptavidin, both of them crucial in ECM–substrate contact [147]. However, molecular models do not consider essential proteins such as cytoskeleton structures, which are relevant for achieving a proper modeling of adhesion.

On the other hand, information regarding adhesion surface modeling is primarily based on kinetics and chemomechanical models. This type of models interrelate the kinetic rates or binding affinities and the mechanical variables, combining two concepts: the mechanics of adhesive tape peeling and the thermodynamics of L-R binding, which relate to kinetic energy of the adhesive bonds [147].

### 6.3. Optimization of Scaffold Design

Computer-aided simulation can be of use in designing scaffold architectures for a wide range of situations, since it is possible to simulate different strain fields and mechanical properties [152].

Broadly, multi-scale modeling utilizes the rigid body model at the system level to estimate the body loading and boundary conditions. Then, an FE analysis of the organ level can estimate the tissue stress [153]. The combination of a system-level model and organ-level model prevents many models from being successful, as they lead to misinterpretation of simulation outcomes. Dao et al. have presented a fully integrated multiscale modeling workflow to account for the interscale synergies, applying the model to a case study on the bone remodeling process of the human jaw [154].

### 6.4. Machine Learning for 3D Printing

The design of scaffolds for tissue engineering can be optimized through three-dimensional printing technologies, which offer unparalleled control over the design of constructs with complex architecture. In particular, extrusion-based 3D printing constitutes a widespread manufacturing technique for scaffold fabrication, due to its low cost and compatibility for processing of a wide range of biomaterials. Optimization of processing parameters, such as speed, pressure, and temperature of the printing process [155], often involves time- and labor-intensive experiments.

In this framework, artificial intelligence (AI) techniques based on machine learning (ML) have arisen to improve 3D printing of materials by analyzing underlying behaviors within a given dataset. The three ML approaches that have been used in these studies are to (1) predict and optimize printing parameters to maximize the structure’s properties, (2) optimize printability of the material, and (3) assess the quality of the prints [156]. For instance, a convolutional neural network trained with a database of hundreds of thousands of geometries from FE analysis was used to design and 3D print composite constructs with superior mechanical properties. In this work, the ML model could identify the geometrical configurations of soft and stiff materials that resulted in the highest toughness and strength when 3D printed as composite structures [157].

### 6.5. Computerized Multiscale Diagnostic System

Bone structure analysis in a patient-specific fashion is often limited to models that incorporate only macro and micro levels. Thus, achieving specific models below the microscale level might lead to significant advances in medical treatment of bone disorders. A conceptual representation of a comprehensive computerized diagnostic system has been proposed by Podshivalov et al. for patient-specific treatment of metabolic bone diseases, fractures, and bone cancer. The diagnostic system is thought to include the following: (a) high-resolution images of human bone at different scales; (b) image processing; (c) 3D reconstruction and multiscale modeling of bone at different structural levels to obtain highly accurate and complete 3D bone models; (d) analysis of bone architecture by applying 3D bone microstructural and topological parameters; (e) multiscale FE analysis of bone structure; and (f) patient-specific scaffolds and/or implants for damaged bone tissue replacement. The system could be also utilized for follow-up and treatment assessment after medication and local therapy procedures [63].

## 7. Current Challenges and Future Directions

To date, numerous biomaterials have been proven to contribute to the development of BTE. However, clinical implementation of these biomaterials is still at its first stages, mainly due to a lack of knowledge about the mechanisms that underlie cell–biomaterial interactions [57]. Particular limitations of current approaches include cell death due to poor nutrient transport in biomaterial scaffolds, inadequate integration of regenerated tissue with the host tissue, and poor mechanical rigidity to provide load-bearing functions [42].

In this context, emerging technologies applied to BTE together with a better understanding of bone biology allow the combination of scaffolds, cells, and growth factors for successful bone repair and regeneration. In order to optimize their implementation, more effective cell isolation, seeding, and culturing methods need to be developed [6]. Moreover, the scaffold properties, discussed in this review, need to mimic those of the native tissue in order to stimulate cell adhesion, migration, proliferation, and differentiation [42].

Overall, the complex scenario present in bone regeneration is extremely dependent on the local microenvironment, including factors such as patient age, injury type, early inflammatory response, ECM composition, physiological adaptation, and angiogenic capacity [61]. One of the concerns that should be addressed is related to adverse effects caused by the presence of biomaterial degradation products for a long period of time. This causes inflammation processes and macrophage polarization from M1 to M2, affecting cytokines such as TNF-α and IL-6, which play a crucial role at the initial stage of the healing process. Thus, further fundamental research should be carried out to describe the mechanisms that control cell–biomaterial interactions [57].

Cell incorporation in BTE scaffolds has been predominantly focused on MSCs. However, induced pluripotent stem cells and embryonic stem cells could be appropriate alternatives to be used for specific disease states [49].

Animal models constitute a major limitation in the preclinical research of effective BTE strategies. Particularly, works on small animals (i.e., mice) have not resulted in relevant results due to major differences in graft size and healing properties. For this reason, it is thought that load-bearing large animal models should be used to assess graft functionality [40]. In this regard, long-bone segmental defects have been modeled in dogs, pigs, goats, and most commonly in sheep, due to their similar body weight to adult humans, well-known mechanical loading, and analogous metabolic and bone remodeling rates [158]. Still, Zeiter et al. [159] performed in 2020 an evaluation of preclinical models in BTE, reporting a lack of standardized and accepted models, and showing the need for optimizing translational aspects (i.e., adaptation to the targeted patient population and justification of the animal model in scientific publications).

Computational approaches might be a useful strategy to optimize and predict substrate performance on bone tissue. However, the translation of computational models from bench to bedside is limited by several issues. First, the determination of patient-specific parameter values becomes difficult when information is limited or of insufficient quality. In addition, current computational models are only validated in animal models, whereas a thorough assessment of the in silico predictive models is critical to encourage health care providers [160]. Furthermore, computational modeling can help improve both scaffold design and the use of high-resolution bioprinting technologies, allowing the design at the cell-specific level at the micro- and nanoscale [42].

Altogether, the future of BTE should integrate fundamental knowledge in bone microenvironment, scaffold design, and simulation and prediction of bone tissue phenomena. Therefore, a multidisciplinary perspective becomes essential for upcoming treatments in bone disorders.

## Figures and Tables

**Figure 1 materials-14-04896-f001:**
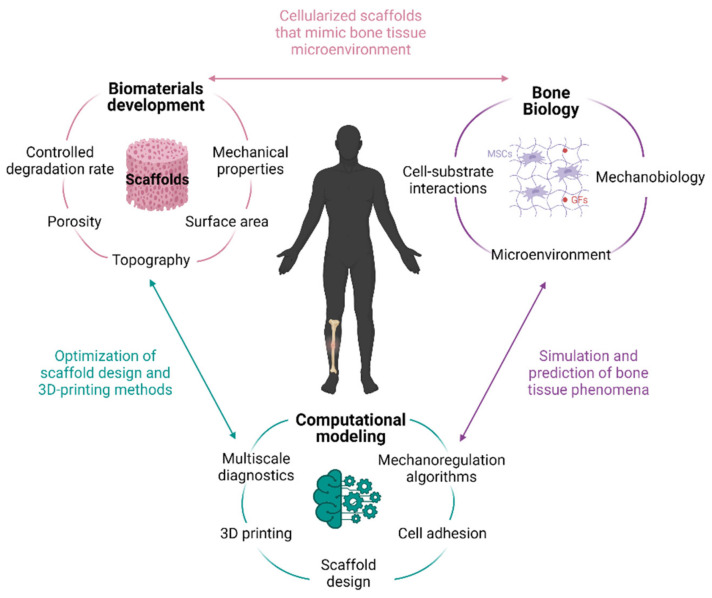
Integration of biomaterials development, bone biology, and computational modeling towards bone tissue engineering.

**Figure 3 materials-14-04896-f003:**
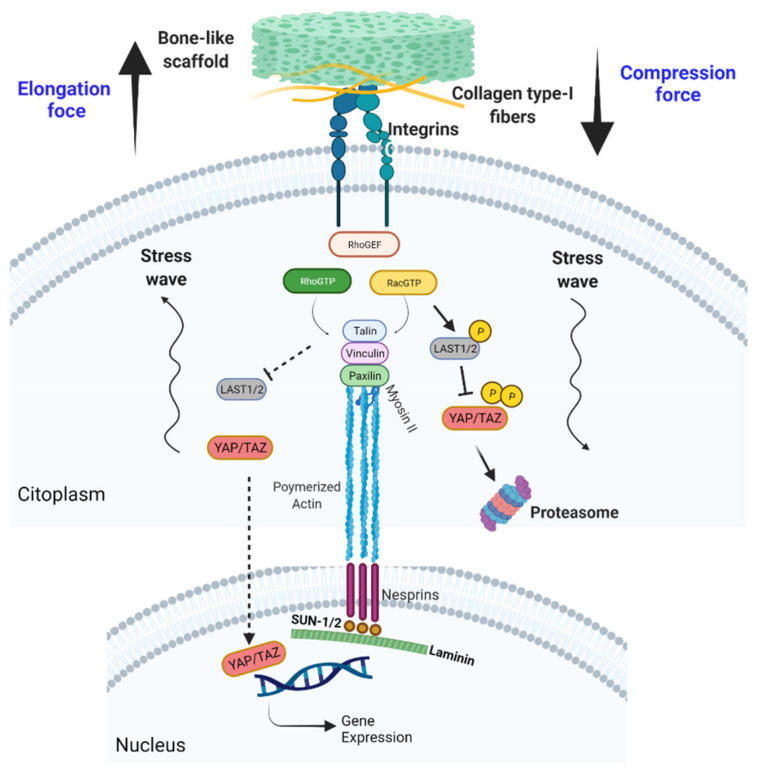
Canonical signaling pathway during cell–ECM interaction with a bone-like scaffold. Mechanical deformation induced by elongation and compression forces triggers recruitment of focal adhesion proteins (vinculin, talin, and paxillin). Focal adhesion kinase (FAK) proteins induce polymerization of actin from the cell cytoskeleton, transmitting the deformation cue to the nucleus cytoskeleton. In parallel, YAP/TAZ molecules are translocated to the nucleus in a mechanical-dependent manner, controlling the expression of genes implicated in proliferation and migration.

## Data Availability

Not applicable.

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
