# Peer review of "A Multidisciplinary Journey towards Bone Tissue Engineering"

_materials, 2021, doi:10.3390/ma14174896_

Round 1

Reviewer 1 Report

This review article is informative and educative in the area. The “cell-biomaterial” and “computational modeling” sections are especially useful for readers. We hope the new technology could help to improve current therapies. The article is well organized and clearly presented.

Author Response

The authors of this review really appreciate the reviewer feedback and want to thank for recognizing the relevance fo the review in the framework of upcoming therapies in BTE.

Reviewer 2 Report

AI implementation is not ideal nor should be presented as such. It strongly depends on knowledge and decision algorithms that are limited.

Author Response

We want to thank the comment. In line 1055, artificial intelligence (AI) is presented as an emerging tool for optimizing biomaterials development. Indeed, it has been already estensively applied in this field. However, it surely presents shortcomings that must be taken into accoi¡unt, and in fact, we talk about the general limitations of computational approaches to reach clinical implementations in line 1122. 

Reviewer 3 Report

The manuscript, "A multidisciplinary journey towards Bone Tissue Engineering" by Pedrero et al. provides an overview of the latest articles published in the field of Bone Tissue Engineering besides describing the fundamentals of bone biology and mechanisms underlying bone remodeling.

1. The manuscript does highlight some of the latest advances that will be useful for researchers in the field of Bone Tissue Engineering, and therefore holds merit for publication. However, in my view, the article is not reflecting a 'multidisciplinary journey' as the authors claim in the title of the manuscript. Figure 1 showing the integration of Chemical Engineering, Bone Biology, and Computational modeling is true and applies the integration of knowledge from 3D scaffold chemistry to Bone tissue engineering. However, it is still a component of Bone tissue engineering characterizing Biomaterials for their effective utilization in aiding bone repair. Likewise, computational modeling and computer-aided simulation for optimization of scaffold design and 3D printing are all considered inevitable components in Bone tissue engineering research for current and as well as future applications. On this basis, the manuscript is not reflecting the title and it requires a change in the title in order to precisely reflect the review area.

2. The efforts that have been made so for to create engineered bone tissue from induced pluripotent stem cells and embryonic stem cells have not been included.

3. Several studies have used sheep as large animal models to create critical-sized bone defects and evaluated the reparative abilities of bone cell types such as Mesenchymal Stem cells (MSCs) on healing bone defects after transplantation using scaffolds. These transplants were also found to enhance bone formation and improve the quality of the regenerating bone tissue compared to other bone grafts. These studies have not been included in the manuscript and need to be included for providing the knowledge gained from large animal models to the readers.

Minor:

Figure 3 - The figure legend mentions (Talin....) which has to be filled in or corrected.

Author Response

Reviewer: The manuscript does highlight some of the latest advances that will be useful for researchers in the field of Bone Tissue Engineering, and therefore holds merit for publication. However, in my view, the article is not reflecting a 'multidisciplinary journey' as the authors claim in the title of the manuscript. Figure 1 showing the integration of Chemical Engineering, Bone Biology, and Computational modeling is true and applies the integration of knowledge from 3D scaffold chemistry to Bone tissue engineering. However, it is still a component of Bone tissue engineering characterizing Biomaterials for their effective utilization in aiding bone repair. Likewise, computational modeling and computer-aided simulation for optimization of scaffold design and 3D printing are all considered inevitable components in Bone tissue engineering research for current and as well as future applications. On this basis, the manuscript is not reflecting the title and it requires a change in the title in order to precisely reflect the review area.

Authors: Firstly, we would like to thank the reviewer for his/her feedback and suggestions. As the reviewer pointed out, the disciplines we are discussing herein (biomaterials development, bone biology and computational modeling) are inevitable components of the domain of bone tissue engineering. In this context, it will be crucial a multidisciplinary team that encompasses knowledge in biology, chemistry, bioinformatic, physic and so on. Thus, instead of addressing or focus predominantly on one of these areas, we present a ‘journey’ through each of BTE components in an attempt to offer an integrative view of the field and. That is why the title refers to a ‘multidisciplinary journey’, which offers a global vision of BTE from each of the disciplines that integrate it.

Reviewer: The efforts that have been made so far to create engineered bone tissue from induced pluripotent stem cells and embryonic stem cells have not been included.

Authors: The utilization of induced pluripotent stem cells and embryonic stem cells was already mentioned in line 655. However, we agree with your assessment that a brief overview on their employment should be included in the manuscript, together with MSCs. That is why we have added a small paragraph on each of them in section 4.3.

Reviewer: Several studies have used sheep as large animal models to create critical-sized bone defects and evaluated the reparative abilities of bone cell types such as Mesenchymal Stem cells (MSCs) on healing bone defects after transplantation using scaffolds. These transplants were also found to enhance bone formation and improve the quality of the regenerating bone tissue compared to other bone grafts. These studies have not been included in the manuscript and need to be included for providing the knowledge gained from large animal models to the readers.

Authors: Thank you very much for your appreciation. The lack of accepted animal models is mentioned in the last chapter of the manuscript as one of the barriers to translational BTE. In the same paragraph (line 1114) a brief note has been added that refers to the use of large animals such as sheep, among others. However, preclinical trials of BTE are beyond the scope of our work. For more extensive information on this subject, there are many references, some already cited in the review, that address this matter more thoroughly.

Minor:

Figure 3 - The figure legend mentions (Talin....) which has to be filled in or corrected.

Authors: Thank you for the minor suggestion, the legend has been properly modified

Reviewer 4 Report

This manuscript is very well-written. It summarizes the approaches to bone tissue engineering (BTE) with abundant background on molecular/biological mechanisms in bone healing and sufficient familiarization to approaches in BTE. I have some minor suggestions.

  1. I don’t think ‘chemical engineering’ is a suitable term to use in Figure 1. ‘Biomaterials development’ would be more inclusive to all the fields involved for scaffolds development.
  2. Table 1 needs a reference column to cite some of the data mentioned.
  3. Section 2.3: Authors should discuss more on other growth factors such as BMP-9 that has found some recent interest due to its excellent osteogenic abilities. Some of these references should be cited, https://doi.org/10.1016/j.msec.2021.112252, https://doi.org/10.1038/s41467-018-08278-4.

Author Response

Reviewer: I don’t think ‘chemical engineering’ is a suitable term to use in Figure 1. ‘Biomaterials development’ would be more inclusive to all the fields involved for scaffolds development.

Authors: The authors do agree with your assessment, the term ‘Biomaterials development’ is indeed a more suitable way to reflect the field of study we are referring to, which in addition to chemical engineering includes basic chemistry and physics and materials engineering, among other disciplines. The figure has been modified accordingly to your feedback.

 Reviewer: Table 1 needs a reference column to cite some of the data mentioned.

Authors: The suggested reference column has been added to table 1.

 Reviewer: Section 2.3: Authors should discuss more on other growth factors such as BMP-9 that has found some recent interest due to its excellent osteogenic abilities. Some of these references should be cited, https://doi.org/10.1016/j.msec.2021.112252, https://doi.org/10.1038/s41467-018-08278-4.

Authors: We want to thank the reviewer indeed for bringing up these interesting works on the use of BMP-9. In section 4.3 we have now included the use of this growth factor as a new strategy for promoting bone formation.

Round 2

Reviewer 3 Report

The manuscript in the present form is reading well and it provides an update in all aspects of the advancements relevant to bone tissue engineering. I recommend the manuscript for publication.